



8       Very Long Period Oscillations in the Atmosphere (0-110 km), Part 2:
9                 Latitude/Longitude comparisons and trends

Dirk Offermann(1), Christoph Kalicinsky(1), Ralf Koppmann(1), and  Johannes Wintel(1,2)
(1)     Institut für Atmosphären - und Umweltforschung, Bergische Universität Wuppertal,  Wuppertal,
18              Germany
(2)     Elementar Analysensysteme GmbH, Langenselbold, Germany
Corresponding author: Dirk Offermann, (offerm@uni-wuppertal.de)
Key Points:  -  oscillations in the period range  5-200 years likely to be self-excited (internal)
-  oscillations very similar at four widely different latitudes and longitudes
-  long-term climate changes difficult to distinguish from long-period oscillations.



Abstract

Atmospheric simulations by computer models exhibit oscillations with multi-annual, decadal, and even centennial periods. These oscillations are especially seen in the temperature. They extend from the ground up to the lower thermosphere. Recent analyses have shown that they exist even if the model boundaries are kept constant with respect to influences of the sun, ocean, and greenhouse gases. Therefore, these parameters appear not responsible for the excitation of these oscillations, However, influences of land surface/vegetation changes had not been enrirely excluded. This is studied in the present analysis. It turns out that such changes are also not candidates for such stimulation. Rather, it appears that the long- period oscillations are excited (internally) in the atmosphere itself.

Long-term trends of atmospheric parameters as the temperature are important for the understanding of the climate change. Their study is mostly  based on data sets that are one to a few decades long. The trend values are generally small, and so are the amplitudes of the long-period oscillations. It can therefore be difficult to disentangle these structures, especially if the interval of trend analysis is comparable to the period of the oscillations. If the oscillations are self-excited, there may be a non-anthropogenic contribution to the climate change which is difficult to determine. Long-term changes of the Cold-Point-Tropopause are analyzed here as an example.

Short Summary

Atmospheric oscillations with periods between 5 and more than 200 years are believed to be self-excited (internal) in the atmosphere, i.e. non-anthropogenic. They are found at all altitudes up to 110 km, and at four very different geographical locations (75°N, 70°E; 75°N,280°E; 50°N,7°E; 50°S,7°E). Therefore, they hint to a global oscillation mode. Their amplitudes are on the order of present day climate trends and it is, therefore, difficult to disentangle them.



## I Introduction

Long-period temperature oscillations have been observed in atmospheric measurements and models (e.g. Meehl et al., 2013; Deser et al., 2014; for further references see Offermann et al. (2021)). The latter authors have reported decadal to even centennial oscillation periods that existed not only at the surface but extended from the ground to the lower thermosphere. It was shown that they were not excited by the sun, the ocean, or greenhouse gases. The amplitudes of these oscillations are not large (i.e fractions of 1 Kelvin). Nevertheless they may be important if long-term trends of temperatures are analyzed, as such trends are on this order of magnitude. Hence, these oscillations may be difficult to disentangle from the trends. This is especially important if the oscillations are part of the internal variability of the atmosphere. Internal and naturally forced variability for instance on decadal time scales is being discussed by Deser (2020) and in the IPCC Climate Change 2021 report (Eyring et al., 2021).

The analyses of Offermann et al (2021) show very long period oscillations that appear to be of internal (self-excited) origin, but whose detailed nature is as yet unknown. Therefore that paper collected a number of characteristic structures that may help to solve that question. This approach is continued here by a comparative study of four locations in the Northern and Southern Hemisphere ( at 50°N vs 50°S , both at 7°E; and at 70°E and 280°E, both at 75°N; coordinates are approximate).

The long-period oscillations of Offermann et al. (2021) were not excited by influences from the sun, ocean, and greenhouse gases. Therefore, self-excitation had been considered as a possibility. However, doubts remained as to a possible excitation by "land-surface"-atmosphere interactions (see their Section 2.2). We therefore compare here locations and occasions with very different surface structures. The location 50°N is in middle of the European land mass. The location 50°S is about 15° south of the tip of South Africa in the Southern ocean. The polar locations are in northernmost Canada and Siberia. Concerning landsurface/atmosphere interaction the locations should behave fairly different. In a further comparison two different seasons (summer/winter) at 50°N, 7°E are considered.

The results of Offermann et al. (2021) had been derived from several atmospheric computer models with special runs whose boundary conditions had been kept constant. In the present analysis we again use two of these: HAMMONIA (38123) and ECHAM6 (for details see that paper). The models showed multi-annual, multi-decadal, and even centennial oscillation periods. These periods were found in a large altitude range, from the ground up to the lower thermosphere. The period values were about constant in this regime. The vertical profiles of oscillation amplitudes and phases, on the contrary, varied substantially. These variations were surprisingly similar for the different oscillation periods. An example of these vertical profiles is shown in Fig.1. The amplitudes vary between maxima and minima. The phases show steps of about 180° which occur at the altitudes of the amplitude minima. For details see Offermann et al. 2021 (their Fig.1). The pronounced vertical structures of the oscillations can possibly help to understand their actual nature.

Long period oscillations may have important influences on the analysis of long-term trends, for instance of temperature. Such trends in the lower and middle atmosphere have been discussed frequently. They are positive or negative, depending on altitude. Recent analyses for the troposphere and stratosphere have been presented, for instance, by Steiner et al. (2020) based on numerous measured data. Such analyses generally cover only a few decades. Therefore, the relative changes are usually small and often comparable to the oscillation amplitudes mentioned. It can sometimes be difficult to analyze them.

Of special interest are temperature changes near the tropopause, as the tropopause is influenced by many parameters and is believed to show a robust "finger print" of climate change (Santer et al., 2004; Pisoft et al., 2021). Tropopause trend analyses have been presented several times (e.g.Zhu et al., 2001; Gettelman et al., 2009; Hu and Vallis, 2019). Long-term changes of tropopause and stratopause altitudes have been analyzed by means of measured and modeled data by Pisoft et al. (2021). They find important changes, such as an increase in tropopause height and a contraction of the stratosphere which they attribute mainly to long-term increases of greenhouse gases. The temperature at the tropopause is frequently studied as the "Cold Point Tropopause" (CPT), i.e. the lowest temperature between troposphere and stratosphere. It is influenced by various atmospheric parameters and therefore discussed as a climate indicator (Hu and Vallis, 2019, Gettelman et al., 2009).





Long term changes of the CPT are of specific interest. They  have been analyzed in the
tropics several times. Zhou et al. (2001) find a negative trend of –0.57±0.06 K/decade in the
time interval 1973-1998. RavindraBabu et al. (2020) find a trend of –1.09 K/decade in the
time interval 2006-2018. Tegtmeier et al. (2020) report trends from –0.3 to –0.6 K/decade
from reanalysis data in the time frame 1979-2005. However, positive trends of tropopause
temperatures have also been discussed (Hu and Vallis, 2019). Positive as well as negative
trends in the range –0.94 to +0.54 K/decade have been reported by Gettelman et al. (2009) in
measured and model data. It is an open question what the reason for these differences and
discrepancies in sign might be.

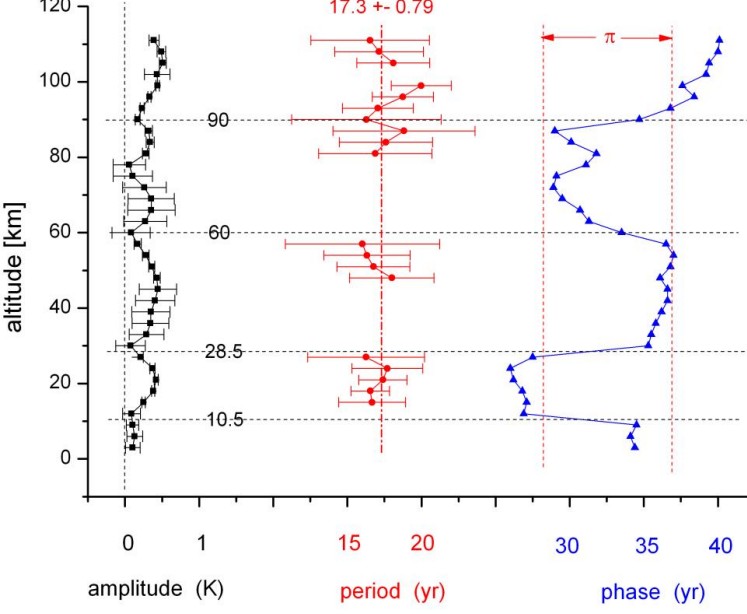

Fig. 1   Vertical structures of long-period oscillations near 17.3 ± 0.8 yr from HAMMONIA
temperatures.
.
The present paper is organized as follows: Section II shows analyses from a HAMMONIA model
run (Hamburg Model of the Neutral and Ionized Atmosphere, 34 years) with fixed boundaries for solar
radiation, ocean, and greenhouse gases. Atmospheric oscillations at northern and southern locations
are compared in terms of their  periods and amplitudes. The periods are between 5 and 28 years.
Section III shows corresponding results from a 400 year long run of the ECHAM6 model
(ECMWF/Hamburg), also with fixed boundaries. Longer periods from 20 to 206 years are analyzed
here. Four locations at different latitudes and longitudes are compared. Section IV discusses the
results. A possible self-excitation of the atmospheric oscillations is considered again.  Furthermore the
implications of the oscillations for the analysis of long-term trends is shown. As an example,  the
behaviour of the Cold Point Tropopause is discussed. Section V summarizes the results.



II   HAMMONIA model

A 34 year run of the HAMMONIA (38123) model has been analyzed for long-period oscillations at Wuppertal (50°N, 7°E). Model details and harmonic oscillation  analysis have been descibed in Offermann et al. (2021).  Model boundaries with respect to the sun, ocean, and greehouse gases were held constant. Nine long-period oscillations with periods between 5 and 28 years have been detected (see Tab.1). They were discussed in terms of self-excited (internal) atmospheric oscillations. Doubts concerning the self-excitation remained , however, because a possible land-surface/ atmosphere interaction could not be excluded. We therefore perform a corresponding analysis here for a conjugate geographic point at 50°S, 7°E. This location is about 15° south of the southernmost tip of South Africa in the middle of the ocean. Hence , the surface/atmosphere interaction should be quite different here from that in the middle of Europe. In case such an interaction plays a role, we hope to see this by comparing various atmospheric parameters. The analysis procedures in the North and the South are exactly the same.

Following  Fig.1 we  study periods and amplitudes of the long-period oscillations. The Figure shows that there are altitude ranges where a period could not be detected. This is attributed to the fact that the oscillation was not excited here, or that it was too strongly damped to be detected (see Offermann et al., 2021). At these altitudes the mean period value of the other altitudes is used as a proxy (vertical dashed red line, $17.3 \pm 0.79$ yr in Fig.1). The proxy is entered into the harmonic analysis and yields estimated values for amplitudes and phases of the oscillation at these altitudes. Details are given by Offermann et al. (2021).

1)  Periods

The above- mentioned nine periods found by Offermann et al. (2021) are repeated  in Tab.1 together with their standard deviations (STD). At 50°S our analysis obtains seven oscillations, that are also shown  in Tab.1.  They all find a correspondence in the northern values. A close agreement is found, that is well within the combined standard deviations in all but one case, and is even within single standard deviation in most cases. These case are indicated by red print in Tab.1.

Table 1 holds a twofold surprise: First, it is interesting to see that long-period oscillations exist in the Southern hemisphere as well as in the Northern hemisphere. Second , it is surprising that the values of the periods are so nearly the same. We would not  expect this if the surface/atmosphere interaction did play a significant role. This is apparently not the case. Our data rather hint to a global oscillation mode that shows up in several periods.

2)  Amplitudes

The vertical amplitude profile in Fig.1 shows a pronounced structure. This offers a valuable tool for our North/South comparison. Offermann et al. (2021)  showed that vertical amplitude profiles of the different oscillations periods were surprisingly similar at the northern location. Their maxima  occurred at about the same altitudes , and so did the minima. (See the accumulated amplitudes in Fig.11 of that Paper.) Hence, the vertical profile of the temperature standard deviation can be used as a proxy for the accumulated amplitude profiles. This is done for the location 50°N, 7°E  (Fig.2, black squares). For the southern location at 50°S, 7°E we also use the temperature standard deviations for a comparison to the North (Fig.2, red dots).





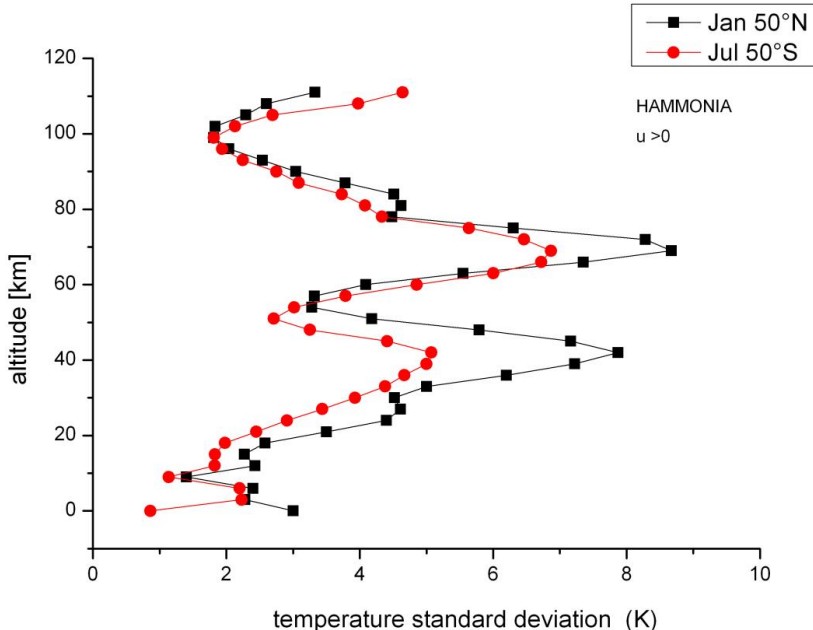

Fig.2   Temperature standard deviations as proxies for oscillation amplitudes in winter. Data   for January at 50°N (black squares) are compared to July at  50° S (red dots).

In the Paper of Offermann et al. (2021)  it was shown that the occurrence of the long-period oscillations was clearly dependent on the direction of the zonal wind: strong oscillation activity was **not** observed for easterly (westward) winds. In the middle atmosphere the zonal wind at solstices is opposite in the Northern and the Southern hemisphere. Hence, comparison of annual mean amplitudes at 50°N and 50°S could be misleading. We  therefore compare here data of the same season: January 50°N to July 50°S ( Fig. 2, zonal wind is eastward), and July 50°N to January 50°S ( Fig.3, zonal wind is westward).



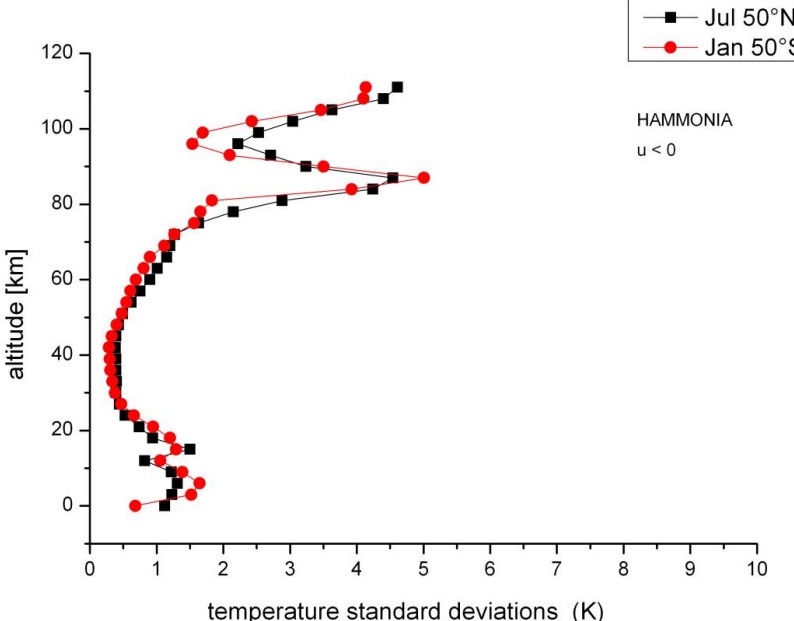

Fig.3   Temperature standard deviations as proxies for oscillation amplitudes in summer. Data are for
July at 50°N (black squares) and for January at 50°S (red dots).
As expected, a comparison of the two pictures shows a large difference of the profiles between
summer and winter at a given latitude, because of the opposite wind directions. The profiles in the
same season, however, are surprisingly similar at 50°N and 50°S. Taking together the  results of
periods and amplitudes it appears that we see essentially the same atmospheric behaviour at 50°N and
50°S. We see no evidence of a possible interaction between the land surface and the atmosphere in the
excitation of the oscillations. We therefore tend to believe that these oscillations are self-excited
(internal).





III  ECHAM6  model
Much longer periods than those in HAMMONIA have been found in the ECHAM6 model (Offermann
et al., 2021). These analyses were based on a 400 year run of that model. Seventeen periods were
observed between 20 years and 206 years (Table 2). They offer further North/South comparisons in
the multi-decadal range and beyond.
1)  Periods
A harmonic analysis of the 400 yr run at 50°S, 7°E is performed in the same way as described in
Offermann et al. (2021) for the North. Sixteen periods can be identified  here, with periods  between
20 years and 16 years.  These are compared to the Northern values in Tab.2.
We find corresponding oscillation values ("pairs") in all cases  except one (206.7 yr in the North).
The last but one column of Tab.2 shows the pair differences, the last column shows the combined
standard deviations. An agreement of periods within the combined standard deviations is found in 12
cases (in red print). In the remaining five cases the periods agree within twice the standard deviations.
This close agreement of the N-S-pairs is similar to that given in Tab.1, and is very remarkable. Again,
there is no evidence of a surface/atmosphere interaction. Together with the HAMMONIA results it
rather  suggests some kind of a three dimensional global oscillation mode.
The HAMMONIA data show substantial differences of oscillation amplitudes between summer and
winter. The oscillation periods of HAMMONIA and ECHAM6 in Tab. 1 and 2 , respectively, are
annual values. As North and South are opposite in season the good agreement of the corresponding
period pairs suggests that seasonal differences of the periods should not be large. We verify this using
the larger set of ECHAM6 data. We compare annual mean oscillation periods to January and July
(mean) values, respectively.
The comparison of the results at 50°N between annual periods (see Tab.2) and corresponding periods
in the January data at 50°N yields 11 coincidences which all agreed within the combined standard
deviations.The corresponding analysis of the annual 50°S data (Tab.2) and the July data at 50°S give
13 coincidences, 12 of which agreed within the combined standard deviations. (One agrees within the
double standard deviations.) Hence,  there is no essential difference between the annual and the
summer and/or winter oscillation periods.
2)  Amplitudes
Amplitudes of the long-period oscillations found in ECHAM6 are analyzed in terms of temperature
standard deviations  as it has been done for the shorter periods of the HAMMONIA model. Also here,
large seasonal differences are expected. Therefore, a North/South comparison is performed  for
corresponding seasons, i.e January North is compared to July South as an example for winter. July
North and January South are compared correspondingly for summer. This is shown in Fig. 4 and 5,
respectively.
Large seasonal differences are seen, indeed, and are similar to those at the shorter periods in Fig. 2
and 3. North and South profiles are, however, very similar if the same seasons are considered, as is
observed for the shorter periods.

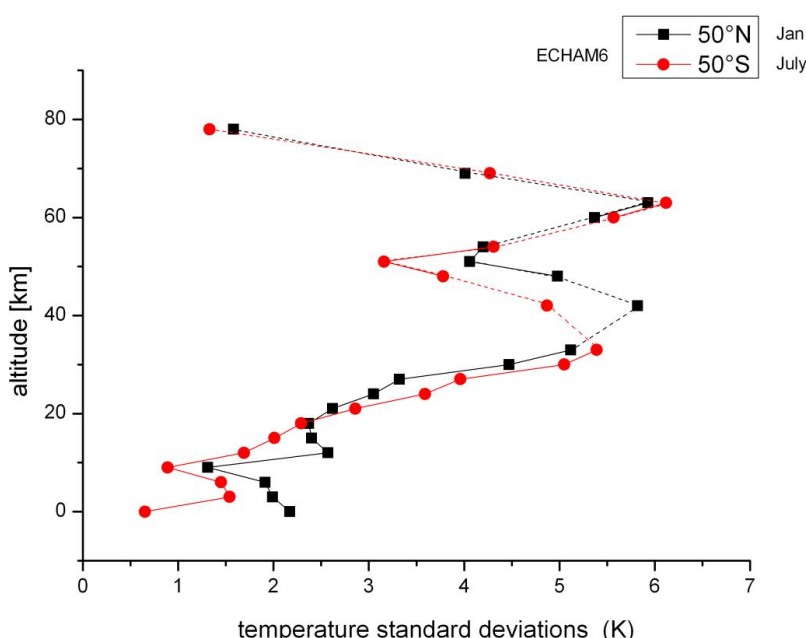

Fig.4   Comparison of ECHAM6 temperature standard deviations in winter.
January 50°N (black squares) and   July 50°S (red dots) are given as examples

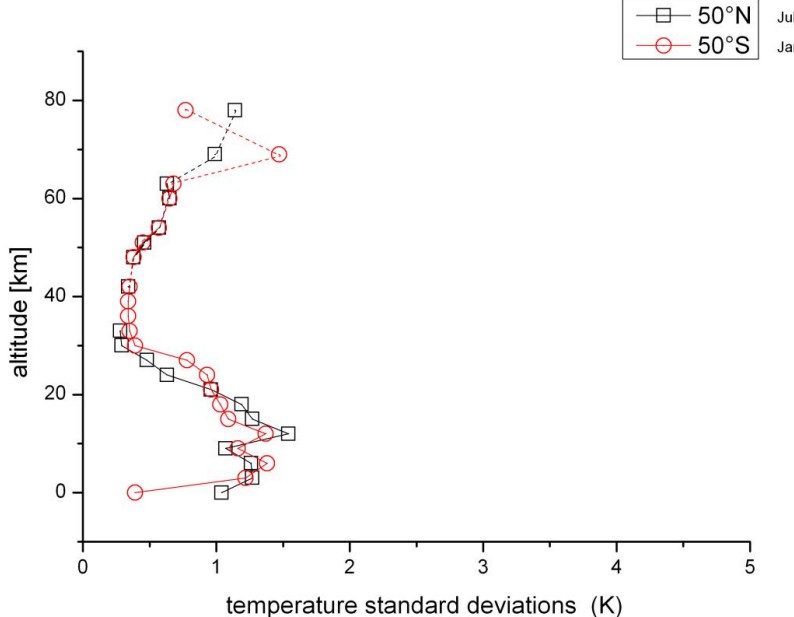

Fig. 5    Comparison of ECHAM6 temperature standard deviations in summer.
July 50°N (black squares) and January 50°S (red circles) are given as examples
<<<<<





The close agreement of the standard deviations at the northern and southern location suggests a
corresponding agreement of the oscillation amplitudes. Such an agreement would be difficult to
understand if the oscillations were excited  by land surface processes. It is rather compatible with a
global oscillation mode self-excited in the atmosphere.
3)   Seasonal Differences
If there is an appreciable influence of land surface/ vegetation on the excitation of the long-period
temperature oscillations in the atmosphere, one would expect a difference of the oscillations in season
at a given location. Such an analysis is in part implicitly contained in the North/South comparisons
given above. We repeat it here in more detail. Oscillation periods in January (northern hemispheric
winter) and July (northern hemispheric summer) are analyzed in the ECHAM6 model at 50°N, 7°E.
Seventeen pairs of oscillation periods can be identified at values similar to those of the annual analysis
shown in the first column of Tab.2. This is shown in Tab.3.  A period near 48 yr could not be found in
July.These results are compared to the annual values of Tab.2. Standard deviations (STD) of the
periods are also given. The second to last column in Tab.3 shows the differences of the periods in
January and July. The last column shows the sum of their standard deviations. A close agreement of
the January and July periods is found: in 14 cases, the periods agree within the combined standard
deviations, which is indicated in red in Tab.3 (in 12 cases even within single standard deviations).  In
three cases, the periods agree within double standard deviations. The agreement of the monthly
periods with the annual ones (first column in Tab.3) is similarly close.
Again, the close agreement of the January and July oscillation periods does not support any
substantial influence of land surface/vegetation on the atmospheric oscillations.
Given the close agreement of the monthly periods, it is interesting to compare their amplitudes.
These are shown in Fig. 6, corresponding to the first column of Fig.1. Accumulated amplitudes are
shown, i.e. the sum of all oscillation amplitudes obtained at a given altitude. The amplitudes could not
be derived for each altitude. Hence, the curves shown in Fig.6 are approximate. The two curves are
quite different. The January curve has high values, is highly structured, and closely resembles in shape
the winter temperature standard deviation profiles in Fig. 4. The values of the  July curve are much
smaller and resemble in shape the summer curves of the standard deviations given in Fig.5. These
agreements again justify the use of temperature standard deviations as proxies of the oscillation
amplitudes.

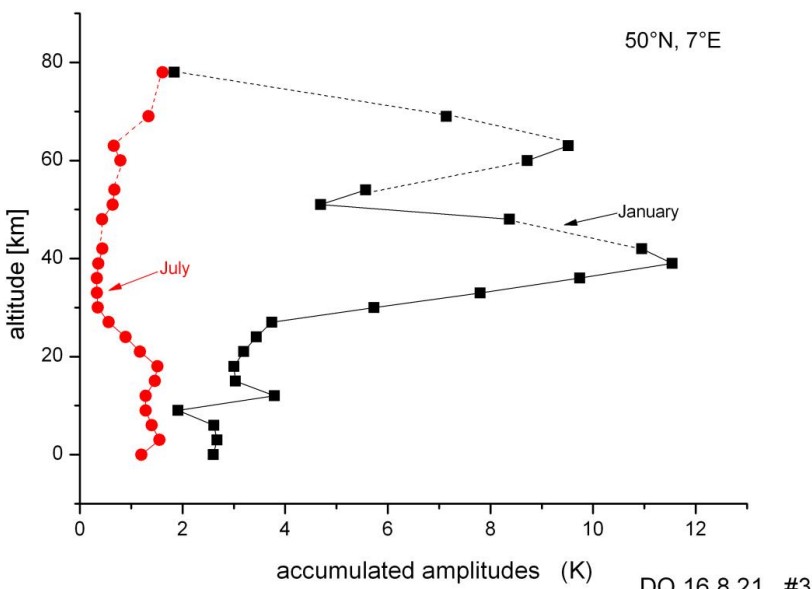

Fig. 6    Long-period temperature oscillations in the ECHAM6 model at 50°N, 7°E. Accumulated
amplitudes are shown vs altitude for the periods given in Tab.3. Black squares are from monthly mean
January data. Red bullets are from July.
The large difference in amplitudes in summer and winter in the stratosphere and mesosphere may be
attributed to the opposite direction of zonal winds in the middle atmosphere in these seasons. It is
surprising that in spite of these large differences the periods of the oscillations are so nearly the same.
This demonstrates that the oscillation period is a robust parameter, as has been discussed by
Offermann et al. (2021).
4)    High Latitudes
Considerable land surface/vegetation differences might also be expected at polar latitudes. We have
therefore analyzed ECHAM6 temperatures  at 75°N, 70°E (Northern Siberia) and 75°N, 280°E
(Northernmost Canada). The two locations are 210° apart in longitude and hence should provide
evidenceof longitudinal structures that may be present. Winter temperatures (January) have been
searched for long period oscillations in the same way as described above. The results are shown in
Tab. 4. For comparison January data at 50°N from Tab.3 are also given. The period differences at the
different locations and the combined standard deviation values have also been calculated  (not shown
here).
The results are quite interesting. The periods found at the two polar locations are very similar.
Seventeen periods have been found at either station, and 16 of these agree within the combined
standard deviations (12 agree even within single standard deviations). The periods at high latitudes are
also quite similar to those at mid latitudes (50°N, 7°E). The 18 periods seen at 50°N find 16
counterparts in either high latitude station. Of these 15  (14) agree within the combined standard
deviations for the 70°E (280°E) station. Eleven periods even agree within single standard deviations in
either case.





Deser et al. (2012) showed in their analysis that the variability of surface temperatures at high
(Northern) latitudes was considerably larger than that at mid and low latitudes. A similar result is
obtained in the present data set for the upper atmosphere. We have caltulated the temperature standard
deviations at the two polar locations (75°N) and show them in Fig. 7. The results at the 70°E and
280°E longitudes are fairly similar. However, as suspected, they are significantly larger than the mid-
latitude values shown in Fig.4. The vertical profile shapes are somewhat different from Fig.4, with the
relative minimum occuring near 30 km at high latitudes as compared to 50 km at mid latitudes.
It was shown above that the standard deviations can be used as a proxy for the (accumulated)
amplitudes of the long period oscillations. This was also verified  and confirmed  for the high latitudes
(not shown here).

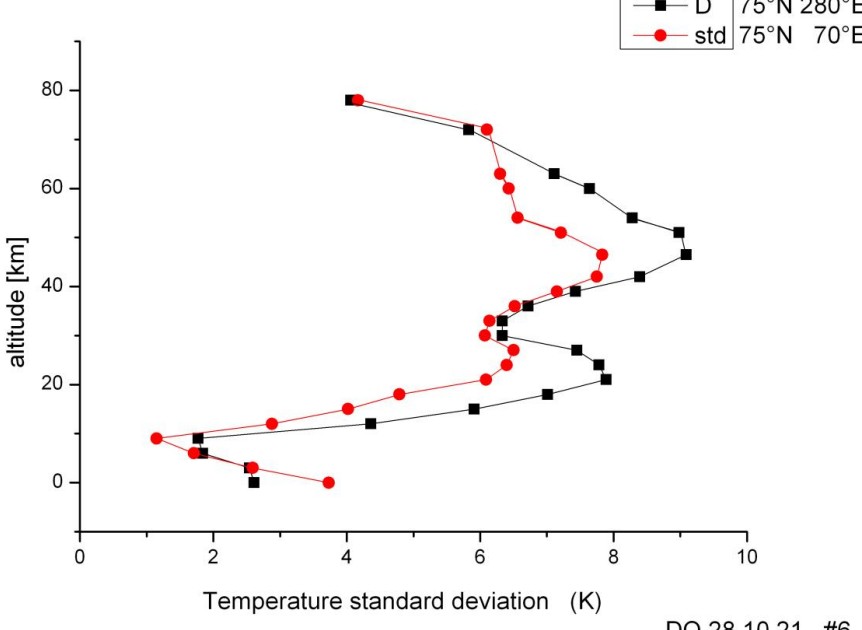

Fig.7   Temperature standard deviations at polar latitudes 75°N, 280°E (black squares) and 75°N,70°E
(red dots) in January



IV   Discussion
1.    Internal oscillations
The boundary conditions of the computer model runs used  by Offermann et al.( 2021) and in the
present analysis  were kept constant. This concerned solar irradiation, the ocean, and greenhouse
gases. Nevertheless, the atmospheres in the models showed pronounced and consistent oscillations. It
was therefore suggested that these oscillations were self-excited or internal in the atmosphere. Land
surface/vegetation changes as external influences, however, were not completely excluded in the
earlier paper. To check such possible influences the models are analyzed here at times and locations
that have different land surface/vegetation conditions. These are on the one hand two corresponding
locations in the Northern and Southern hemisphere (50°North and South at 7°East). On the other hand
two different seasons are compared at the same location (50°North, 7°East).  Finally, two polar
locations at very different longitudes are studied (75°N at 70°E and 280°E, respectively).
The results for all northern and southern locations are very similar. This concerns above all the
oscillation periods. A large number of pairs of oscillation at the different locations with very similar
periods is obtained.  Also the amplitudes  are found to be similar when comparing the corresponding
seasons. Furthermore, comparison of the two different seasons (summer/winter) at the same location
shows very similar periods. This is surprising because  the amplitudes are very different. We  conclude
from these various results that it is unlikely that the long-period oscillations originate from land
surface/vegetation processes! They rather appear to be self-excited as mentioned.
The large summer/winter difference in amplitudes (standard deviations) applies to one pair of
North/South locations (50°N/S, 7°E). The global analyses of Deser et al. (2012) indicate , however,
that this may be a global phenomenon (Deser et al., 2012, their Fig.16). This is seen if their December-
January data are compared to our January data: Northern values are much larger than Southern values.
It thus appears that our North/South difference is part of an extended (global) structure.
However, in July their and our values disagree: they do not see much difference between 50°N and
50°S, whereas here in Fig 2-5 the Northern values are much smaller than those in the South.
This discrepancy may find its explanation in the vertical structure of the data. The data of Deser et
al. are bottom temperatures. Our data , on the other hand, cover the whole altitude range up to the
lower thermosphere . However, at the lowest altitude (surface) all of our Southern amplitudes (given
as standard deviations) are much smaller than their Northern counterparts (Fig. 2-5). This is the case
even though the altitude profiles are otherwise very similar. It is interesting  that this difference is
limited to the lowermost altitude , and  disappears at the next higher altitude level (3 km). This applies
to the two different models HAMMONIA as well as ECHAM6. The difference of the two lowermost
levels is significant as the statistical error of the standard deviations is 12% for HAMMONIA and
3.5% for ECHAM6.
A quantitative analysis of the two models at the lowest altitudes (50° N or S) in Fig. 2-5 shows that
the January values are high in the North (2.2-3.0 K) and small in the South (0.39-0.68 K). Contrary to
this, the July values are comparatively low as well in the North (1.04-1.12 K) as in the South (0.65-
0.86 K). This is very similar to the results of Deser et al. (2012).  Therefore, special care obviously
needs to be taken when comparing climatological surface parameters of the North to the South, and to
higher altitudes.
Internal variability in the atmosphere has been discussed several times in the literature (e.g. Deser
(2020) and references therein). This is thought to be caused by the chaotic dynamics of the atmosphere
and oceans, and to be generally unpredictable more than a few years ahead of time. It remains to be
determined how this is related to our internal oscillations.





2.   Implications of internal oscillations
a)    Temperature trends
New long-term temperature trends in the troposphere and stratosphere have recently been presented
by Steiner et al. ( 2020). Data cover about four decades (1980 – 2020). These authors find trends on
the order of -0.2 K/decade in the lower stratosphere  (near- global averages, their Fig. 8). For
comparison, we show ECHAM6 data for 50° N, 7°E at 18 km altitude in our Fig.8. These data are
annual mean residues, i.e. the mean value has been subtracted from the annual data set.  The series has
been smoothed by a 16 point running mean. The Figure shows trend-like increases or decreases of 0.2
K/dec or even steeper over 4 decade intervals.This is indicated by the slant red lines that give an
increase of 0.2 K/dec.  This variability of the ECHAM6 data is obviously of internal origin because we
use model runs with fixed boundaries also here.
The comparison with  Steiner et al. (2020) is approximate because our data are local (50°N, 7°E),
whereas Steiner et al. give global means. Such means tend to smooth all variability to some extent.
Nevertheless, the results suggest that the long-term trends derived by Steiner et al. (2020) may contain
some contribution of internal (i.e. non-anthropogenic) variability. This confirms a corresponding result
of  these authors saying that "…there may be a nonnegligible internally generated component to the
larger stratospheric trends…" (see their Section 5).

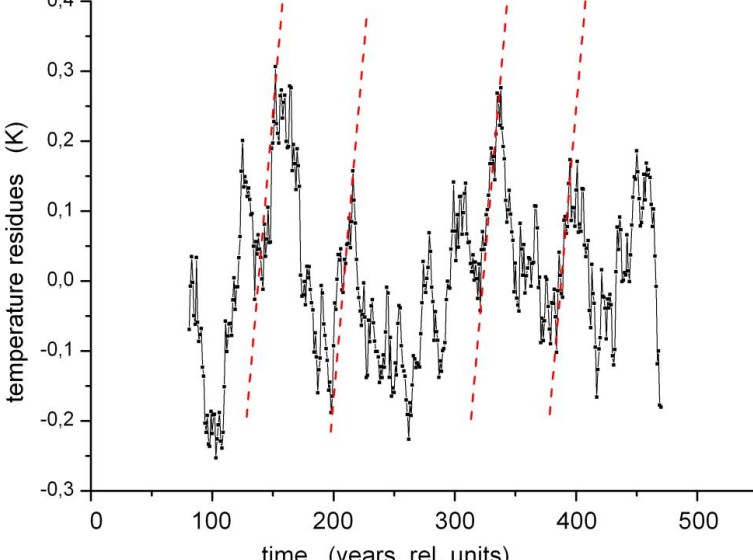

Fig.8   ECHAM6 annual temperature residues at 50°N, 7°E, 18 km altitude. Data have been smoothed
by a 16 point running mean. Time is in relative units. Inclined dashed (red) lines have a gradient of 0.2
K/decade.





b)    Cold Point Tropopause.

The Cold Point Tropopause (CPT) is  frequently discussed as a climate indicator ( see e.g. Hu and
Vallis, 2019; Gettelman et al., 2009). A similar parameter is the Lapse Rate Tropopause (LRT), which
we do not discuss here as it is generally close to and behaves similarly as the CPT (Pan et al., 2018;
RavindraBabu et al., 2020).
We  analyze long-term changes of the Cold Point Tropopause (CPT) in the ECHAM6 model   at
50°N, 7°E and the corresponding Southern Hemisphere location (50°S, 7°E) as part of our
North/South comparison. The lowest temperatures are found in this model at 11.5 km (208.67 hPa)
and 12.4 km (181.16 hPa) (this is the altitude resolution of the data). We have selected the lowest
temperature at these two altitudes and thus formed a data set that approximates the Cold Point
Tropopause, considering our limited altitude resolution.
The results are shown in Fig.9. The figure compares our CPT data at the two locations. To study data
that are corresponding,  winter values are shown, i.e January data in the Northern hemisphere and July
data in the Southern hemisphere.  The data have been smoothed by a 16 point running mean to
suppress the short term variability that is large (5 K pp).  The picture shows that the Southern CPT are
somewhat lower than the Northern ones. Most interesting is the strong variability in either data set,
including  some apparent periodicity. The latter is indicated by the vertical dashed lines at 60 year
intervals. On time scales of  decades, positive and negative trends are seen. The positive trends are
comparable to the dashed (blue) straight lines that have a gradient of 1 K/dec. The picture shows that
such gradients or even steeper ones are not uncommon in the data. The decreasing branches show
similar ( negative) gradients.

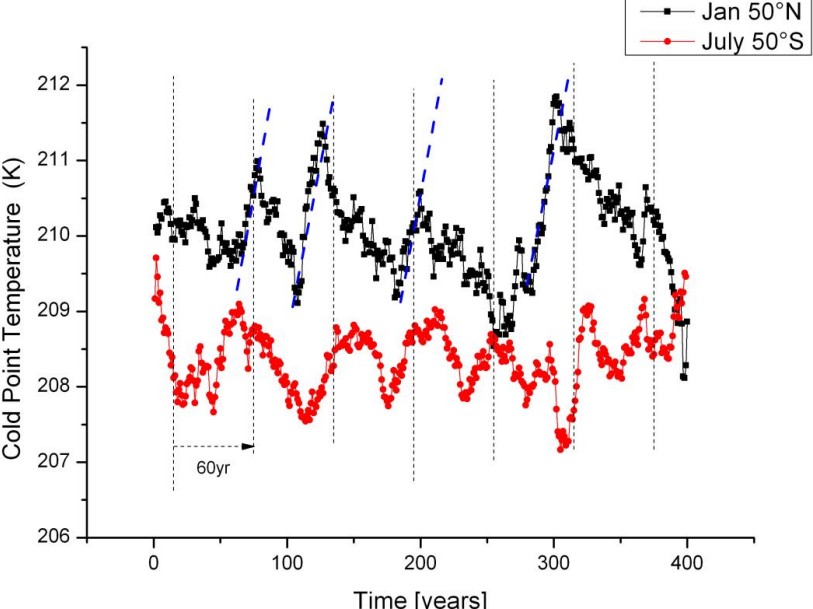

Fig.9    Cold Point Tropopause temperatures in ECHAM6.
Winter data are shown for 50°N, January (black) and 50°S, July (red). Dotted vertical lines (black)
indicate a 60 yr  periodicity.  Inclined dashed lines (blue) show a trend of 1 K per decade. Time is in
relative units.



Gradients on this order of magnitude are given in the literature. Amazingly, positive as well as
negative values are found, as reported in Section I. Figure 9 shows that this may not be surprising, but
may occur quite naturally depending on the time interval chosen for the trend determination. The
quasi-periodic behaviour of the CPT plays a role here and suggests a possible connection to the
internal oscillations of the atmosphere.
We therefore perform harmonic analyses of the CPT data similarly as described above for annual
temperatures in Tab.2. The CPT data are monthly data of January and July, respectively. It was shown
above that there is little difference between annual and monthly oscillation periods, and it was checked
that this applies here, too.
The harmonic analyses of the data yield a number of internal oscillation periods in the period range
of Tab.2, indeed. The results at the Northern and Southern locations are compared in Tab.5. The table
shows that the periods in the North and South form pairs similarly as in Tab.1 and 2.   Eleven
coincidences are obtained. Seven of these agree within the combined standard deviations (red in the
last two columns of Tab.5). Four agree within the double standard deviations (black in Tab.5). All
periods listed in Tab.5 also  find a counterpart in the corresponding (North or South) columns of
Tab.2. Also, these pairs agree within combined standard deviations (except one). It thus appears that
the Cold Point Tropopause is at least partly controlled by the internal atmospheric oscillations. This
applies to the North as well as to the South, i.e. the North/South symmetry shown above is also found
in this parameter.
The amplitudes of the CPT oscillations are found  quite variable (not shown here). The Northern and
the Southern data both show strong amplitude peaks near 60 years. This fits to the data shown in Fig.8.
Low frequency oscillations (LFO) in the multi-decadal range (50-80 years) have frequently been
discussed for surface temperatures. They have, for instance, been interpreted as internal Atlantic
Multidecadal Variability or Pacific Decadal Oscillations/Interdecadal Pacific Oscillations (e.g. Meehl
et al., 2013, 2016; Lu et al., 2014; Deser et al, 2014; Dai et al., 2015).   It appears that  internal
oscillations play a role here as contributors to the CPT variations in either hemisphere. Great caution is
therefore advised when interpreting tropopause changes in the context of the anthropogenic  long term
climate changes (e.g. Pisoft et al., 2021).

V   Summary and Conclusions
1)  Self-excitation of oscillations
Present day sophisticated atmospheric computer models exhibit long period temperature oscillations
in the multi-annual, decadal, and even centennial year  range. Such oscillations may be found even if
the model boundaries are kept constant concerning the influences of solar radiation, the ocean, and the
variations of greenhouse gases (Offermann et al., 2021). A possible influence of land surface/
vegetation changes, however, was undecided yet. Therefore, in the present analysis oscillation periods
are compared at locations/occasions  with different land surface/vegetation behaviour, hoping to see
possible differences in oscillation periods. Two cases are studied: First, a location in the Northern
hemisphere (50°N, 7°E) and its counterpart in the Southern hemisphere (50°S, 7°E) are considered.
The Northern location is in the middle of Europe, whereas the Southern location is 15° south of the tip
of South Africa in the middle of the Southern ocean. Alsao, two different seasons are compared in the
Northern location (January and July). Two models are studied (HAMMONIA, ECHAM6) for medium
and long oscillation periods (5 to beyond 200 years). Second, two polar latitude locations are studied
at 75°N, 280°E and 75°N, 70°E. Their land surface/vegetation conditions are quite different from the
other locations. Interestingly, the periods obtained for the contrasting cases are all found very similar.
It is therefore concluded that the oscillations very likely are internally excited in the atmosphere.
2)  Robust periods
Oscillation periods were found to be very similar in three different atmospheric models (Offermann et
al., 2021). It was thus concluded that the period is a very robust parameter. This is confirmed in the
present analysis. Amplitudes are found quite different in contrasting seasons (January/July), with
winter values much larger than summer values in the middle atmosphere. The periods, however, are
about the same.
3)  Global oscillation mode
Long period oscillations were analyzed at four locations quite different in latitudes and longitudes.
Their periods are found surprisingly similar. A given oscillation period is found in a similar way at
many/all altitudes from the ground up to the lower thermosphere. The altitude distribution is about a
straight line. This is not the case for the amplitudes. The respective profiles are highly structured,
especially in winter. However, the profiles at different locations  are fairly similar if corresponding
seasons (summer/winter) are compared. This result and the similarity of the oscillation periods hint to
a three-dimensional global oscillation mode. To substantiate this, a more extended  global analysis is
suggested for the future.
4)  Trends and long periods
Long- term trends in atmospheric parameters are frequently analyzed in the context of the ongoing
climate change. Trend values are mostly small, and it is sometimes difficult to determine whether or to
what extent they are anthropogenic in nature. In this context internal oscillations can play a role even
if their amplitudes are small. If the oscillation period is on the order of the interval used for the trend
analysis it may become difficult to disentangle trend and oscillation.
As an example the Cold Point Tropopause (CPT) in the 400 year run of the ECHAM6 model with
fixed boundaries is analyzed at  two North/South locations. Strong trend-like increases or decreases of
CPT values are seen on decadal time scales. They are on the order of the trend values discussed in the
literature. They are, however, not of anthropogenic origin, as is frequently assumed in the literature.





Harmonic analysis of the CPT values yields oscillation periods that are very similar for the North and
South location,  and are similar to the values otherwise given in this analysis. Apparently these internal
oscillations are important contributors to the CPT variations observed.





Author Contribution
DO performed the data analysis and prepared the manuscript with the help of all co-authores.
JW managed the data collection and preparation.
ChK helped with the geographical analysis.
R.K provided interpretation and editing the manuscript.
Competing Interests
The authors declare that they have no conflict of interest.





Acknowledgement

We thank Hauke Schmidt (MPI Meteorology , Hamburg, Germany) for many helpful discussions.
HAMMONIA ans ECHAM6 simulations were performed at and supported by German Climate
Computing Centre (DKRZ) and are greatfully achnowledged.
This work was done within the CHIARA (CHaracterisation of the Internal vARiability of the
Atmosphere) project as part of the ISOVIC (Impact of SOlar, Volcanic and Internal variability on
Climate) project in the framework of the ROMIC II program (Role of the Middle Atmosphere in
Climate).The project was financially supported by the Federal Minstry for Education and Research
within the ROMIC II program under grant no. 01LG1909A.



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



Table 1    Oscillation periods and their standard deviations at 50°N, 7°E vs 50°S, 7°E  (HAMMONIA
model)

| | Period (yr) 50°N | STD | Period (yr) 50°S | STD | difference of periods | combined STD |
|---|---|---|---|---|---|---|
| 1 | 5,34 | ± 0,1 | 5,61± | 0,15 | -0.27 | 0.25 |
| 2 | 6,56 | 0,24 | | | | |
| 3 | 7,76 | 0,29 | 7,42 | 0,28 | 0.34 | 0.57 |
| 4 | 9,21 | 0,53 | 9,24 | 0,45 | -0.03 | 0.98 |
| 5 | 10,8 | 0,34 | 10,7 | 0,18 | 0.1 | 0.52 |
| 6 | 13,4 | 0,68 | 13,2 | 0,86 | 0.2 | 1.54 |
| 7 | 17,3 | 1,05 | 16,5 | 1,3 | 0.8 | 2.35 |
| 8 | 22,8 | 1,27 | -- | -- | | |
| 9 | 28,5 | 1,63 | 30,3 | 4,6 | -1.8 | 6.23 |

Table 2   Oscillation periods and their standard deviations at 50°N, 7°E  vs  50°S, 7°E (ECHAM6
model)
980 .

| | Period (yr) 50°N | STD | Period (yr) 51°S | STD | difference of periods | combined STD |
|---|---|---|---|---|---|---|
| 1 | 20 | ±0,35 | 20,1 | ±0,4 | -0,1 | 0,75 |
| 2 | 20,9 | 0,15 | 21,8 | 0,37 | -0,9 | 0,52 |
| 3 | 22,1 | 0,23 | 23,2 | 0,33 | -1,1 | 0,56 |
| 4 | 23,8 | 0,42 | 24,3 | 0,41 | -0,5 | 0,83 |
| 5 | 25,3 | 0,46 | 26,1 | 0,44 | -0,8 | 0,9 |
| 6 | 27,3 | 0,41 | 28,6 | 0,44 | -1,3 | 0,85 |
| 7 | 30,2 | 0,49 | 31,8 | 0,58 | -1,6 | 1,07 |
| 8 | 33,3 | 0,84 | 34,5 | 0,58 | -1,2 | 1,42 |
| 9 | 36,9 | 1,17 | 38,3 | 1,05 | -1,4 | 2,22 |
| 10 | 41,4 | 0,97 | 43 | 1,52 | -1,6 | 2,49 |
| 11 | 48,4 | 1,73 | 49,7 | 1,78 | -1,3 | 3,51 |
| 12 | 58,3 | 1,77 | 60,3 | 2,33 | -2 | 4,1 |
| 13 | 64.9 | 2.98 | 66.5 | 2.5 | -1.6 | 5.48 |
| 14 | 77.5 | 3.94 | 84.8 | 4.74 | -7.3 | 8.68 |
| 15 | 95.5 | 5.86 | 110.9 | 10.9 | -15.4 | 16.76 |
| 16 | 129.4 | 14.5 | 160.2 | 8.88 | -30.8 | 23.38 |
| 17 | 206.7 | 16.3 | | | | |






Table 3   Temperature oscillation periods (yr) at 50°N,7°E,  standard deviations (std), and column
differences

| | Period Annual | STD | Period January | STD | Period July | STD | difference Jan-July | STD sum Jan+July |
|---|---|---|---|---|---|---|---|---|
| 1 | 20 | 0,35 | 19,6 | 0,33 | 19,8 | 0,52 | -0,2 | 0,85 |
| 2 | 20,9 | 0,15 | 20,8 | 0,32 | 21 | 0,18 | -0,2 | 0,5 |
| 3 | 22,1 | 0,23 | 22,4 | 0,33 | 22,2 | 0,38 | 0,2 | 0,71 |
| 4 | 23,8 | 0,42 | 24,1 | 0,19 | 24,1 | 0,31 | 0 | 0,5 |
| 5 | 25,3 | 0,46 | 25,3 | 0,49 | 26,1 | 0,21 | -0,8 | 0,7 |
| 6 | 27,3 | 0,41 | 27,8 | 0,76 | 27,7 | 0,17 | 0,1 | 0,93 |
| 7 | 30,2 | 0,49 | 30,3 | 0,62 | 30,2 | 0,76 | 0,1 | 1,38 |
| 8 | 33,3 | 0,84 | 33,1 | 1,03 | 33,7 | 0,55 | -0,6 | 1,58 |
| 9 | 36,9 | 1,17 | 37,5 | 1,05 | 38,1 | 1,3 | -0,6 | 2,35 |
| 10 | 41,4 | 0,97 | 41,5 | 1,49 | 44,3 | 1,23 | -2,8 | 2,72 |
| 11 | 48,4 | 1,73 | 48,3 | 1,69 | -- | -- | -- | -- |
| 12 | 58,3 | 1,77 | 57,9 | 0,53 | 53,3 | 1,77 | 4,6 | 2,3 |
| 13 | 64,9 | 2,98 | 63,5 | 2,7 | 66,2 | 1,92 | -2,7 | 4,62 |
| 14 | 77,5 | 3,94 | 77,1 | 2,5 | 79,1 | 5,11 | -2 | 7,61 |
| 15 | 95,5 | 5,86 | 97,6 | 7,81 | 103,8 | 5,4 | -6,2 | 13,21 |
| 16 | 129,4 | 14,5 | 130,1 | 9,03 | 121,1 | 9,32 | 9 | 18,35 |
| 17 | 206,7 | 16,3 | 169,3 | 10,55 | 183,4 | 7,51 | -14,1 | 18,06 |
| 18 | -- | -- | 239 | 15,3 | 216,2 | 14,67 | 22,8 | 29,97 |



Table 4 Temperature oscillation periods (yr) and their standard deviations (STD) at 50°N, 7°E; 75°N, 70°E; and 75°N, 280°E in January.

|  | 50°N, 7°E | STD | 75°N, 70°E | STD | 75°N, 280°E | STD |
|---|---|---|---|---|---|---|
| 1 | 19.6 | 0.33 | 19.6 | 0.44 | 19.2 | 0.26 |
| 2 | 20.8 | 0.32 | 21 | 0.19 | 20.7 | 0.32 |
| 3 | 22.4 | 0.33 | 22.8 | 0.4 | 22.6 | 0.32 |
| 4 | 24.1 | 0.19 | 24.4 | 0.2 | 24.4 | 0.3 |
| 5 | 25.3 | 0.49 | 25.8 | 0.55 | 25.3 | 0.27 |
| 6 | 27.8 | 0.76 | 28.9 | 0.34 | 26.7 | 0.29 |
| 7 | 30.3 | 0.62 | 30.9 | 0.66 | 29.9 | 0.7 |
| 8 | 33.1 | 1.03 | 33.1 | 0.51 | 32.6 | 0.69 |
| 9 | 37.5 | 1.05 | 35.8 | 0.93 | 37 | 0.6 |
| 10 | 41.5 | 1.49 | 40.5 | 0.9 | 39.7 | 0.8 |
| 11 |  |  | 44.7 | 1,25 | 43.9 | 1.29 |
| 12 | 48.3 | 1.69 | 51.1 | 2.22 | 50.9 | 2.49 |
| 13 | 57.9 | 0.53 |  |  |  |  |
| 14 | 63.5 | 2.7 | 61.4 | 1.75 | 64.4 | 2.73 |
| 15 | 77.1 | 2.5 | 76.7 | 4.04 | 82.2 | 2.16 |
| 16 | 97.6 | 7.81 | 95.8 | 5.97 | 91.2 | 5.91 |
| 17 | 130.1 | 9.03 | 149.4 | 9.95 | 139.4 | 10.99 |
| 18 | 169.3 | 10.55 |  |  |  |  |
| 19 | 239 | 15.3 | 232.5 | 13.1 | 244.5 | 22.8 |





Table 5   Cold Point Tropopause oscillations in winter at 50°N and 51°S, standard deviations, and
column differences

| | CPT period (yr) Jan 50°N | STD | CPT period (yr) July 51°S | STD | difference of periods | combined STD |
|---|---|---|---|---|---|---|
| 1 | 19.8 | 0.27 | 20.2 | 0.56 | -0.4 | 0.83 |
| 2 | 21.1 | 0.44 | 22.2 | 0.38 | -1.1 | 0.82 |
| 3 | 24.9 | 0.32 | 24.1 | 0.38 | 0.8 | 0.7 |
| 4 | 28.8 | 1.26 | 26.2 | 0.32 | 2.6 | 1.58 |
| 5 | 31.3 | 1.84 | 32.8 | 0.6 | -1.5 | 2.44 |
| 6 | 42.3 | 1.64 | 39.8 | 1.33 | 2.5 | 2.97 |
| 7 | 48.3 | 3.22 | 47.1 | 3.22 | 1.2 | 6.44 |
| 8 | 58 | 2.22 | 65.5 | 2.14 | -7.5 | 4.36 |
| 9 | 75.1 | 4.45 | 81.8 | 5.6 | -6.7 | 10.05 |
| 10 | 107.7 | 6.64 | 96.4 | 8.7 | 11.3 | 15.34 |
| 11 | 179.3 | 13.3 | 171.5 | 21.7 | 7.8 | 35 |
