# Peer review of "3 4 5 Pap11 DO06023 pdf2 Very Long Period Oscillations in the Atmosphere (0-110 km), Part 2: Latitude/Longitude comparisons and trends Dirk Offermann(1), Christoph Kalicinsky(1), Ralf Koppmann(1), and Johannes Wintel(1,2) 18 Institut für Atmosphären - und Umweltforschung, Bergische Universität Wup"

_Atmospheric Chemistry and Physics, 2022_

## Author Response (AR1)

DO 10.2.23

Authors Response summary

- **RC1**: , Anonymous Referee #1, 15 Nov 2022

  This is interesting paper, which confirms previous results of authors. It shows as a new result that the last potential external source, impact of land surface/vegetation also does not have any substantial influence on the observed internal atmospheric oscillations in the whole interval of altitudes 0 – 80(110) km. However, the paper requires some improvements, I would say moderate revision - see comments below.

  Comments:

  Tables 1 and 2. Periods in these Tables differ to some extent. For $50^{o}$N:  Hammonia 28.5 years corresponds to ECHAM6 periods 27.3 and 30.2, Hammonia period 22.8 corresponds ECHAM6 periods 22.1 and 23.8, ECHAM6 periods 20.9 and 20 do not have HAMMONIA counterparts. Have you some explanation for these differences? Make a comment on that in the paper. Would it be possible to get some shorter ECHAM6 periods for comparison with HAMMONIA?

  Table 3: Period 206.7 years should be moved from line 17 to line 18.

  Figures 2 and 4: Main peaks in 0-80 km, located near 70 and 40 km, occur in both Figures. You can use it as another supporting evidence for your results.

  Figure 7: Why the profiles in Fig. 7 differ so clearly from those in Fig. 4. They differ not only in the different positions of minimums at 30 versus 50 km but also by presence/absence of maximum near 70 km. Can you make a broader comment in the paper about this difference. Figure itself – I recommend remove "D" and "std" (in fact both are std).

  Summertime profiles are strongly damped compared to wintertime profiles due to direction of zonal wind. Does it mean/indicate that the observed oscillations propagate from below?

  Why the profiles are so structured? Does it mean/indicate in-situ generation in individual layers?

  Line 676: Could you specify typical time scales? This might be important for long-term trend determination.

  References: Steiner et al. (2020) is missing in the list of references. Pisoft et al. – is it in press (list of references) or (2021) (in the text)?

  Wording and misprints:

    o   301: "16" should be "160" !!

**Citation**: https://doi.org/10.5194/acp-2022-677-RC1

- **AC1**
o  Response to comments of Referee 1 (RC1):

ACP- 2022-677   Very Long Period Oscillations in the Atmosphere (0-110 km), Part 2: Latitude/longitude comparisons and trends

D.Offermann, Ch.Kalicinsky, R.Koppmann, J.Wintel

We thank the Referee for his careful and detailed comments! We hope to have responded to them below to his satisfaction. Changes made are indicated in red in the text.

Comment #1: "Tables 1 and 2. Periods in these Tables differ to some extent. For 50°N: Hammonia 28.5 years corresponds to ECHAM6 periods 27.3 and 30.2. Hammoniia period 22.8 corresponds to ECHAM6 period 22.1 and 23.8, ECHAM6 periods 20.9 and 20 do not have Hammonia counterparts. Have you some explanation for thes differences? Make a comment on that in the paper. Would it be possible to get some shorter ECHAM6 periods for comparison with Hammonia?"

Response:

The periods cited for Hammonia and ECHAM6 agree within their (combined) error bars.

   Missing periods (counterparts) "occur" at several places in Tab.1 –4. We believe that in these cases the amplitudes were too small to detect the periods.

A corresponding comment had been given in Paragraph 2 of Section II and has again been added in "Section III (ECHAM6 model)", "1) Periods" (paragraph 1).

Shorter ECHAM6 periods were not shown in Part 1 (ACP 2021) and Part 2 (this paper) of the paper as the emphasis is on the long and very long periods.

Comment #2: "Table 3: Period 206.7 years should be moved from line 17 to line 18."

Response: Done as suggested, Thank you!

Comment #3: "Figures 2 and4: Main peaks in 0-80 km , located near 70 km and 40 km , occur in both Figures. You can use it as another supporting evidence for your results."

Response: Thank you! A corresponding comment was added in "Section III, 2)Amplitudes" paragraph 2.

Comment #4: "Figure 7: Why the profiles in Fig. 7 differ so clearly from those in Fig.4. They differ not only in the different positions of minimums at 30 versus 50 km but also by presence/absence of maximum near 70 km. Can you make a broader comment in the paper about this difference. Figure itself- I recommend remove "D" and "std" (in fact both are std).

Response: This is a complicated question, that is presently under analysis: The vertical profile of amplitudes or standard deviations as shown in Fig.4 and

6 can be calculated for all periods shown in Tables 1-4. At mid latitudes (50°N) they all show about the same profile form as in these Figures. Surprisingly this is not the case at the higher latitudes (75°N). Here the profiles can be very different. They depend as well on the oscillation period as on the geographic longitude (70°E vs 280°E). This leads to the smeared profile structure and the longitudinal differences in the averaged profiles seen in Fig. 7. The reason for these differences is presently unknown. It must have to do with- and should eventually lead to the understanding of- the excitation mechanism of the long-period oscillations. This is, however, beyond the scope of the present paper.

   A corresponding comment has been added as the last paragraph of Section III.

   "D and std" have been removed as suggested.

Comment #5:   Summertime profiles are strongly damped compared to winter time profile due to direction of zonal wind. Does it mean/indicate that the observed oscillations propagate from below?

Response:   The nature of our oscillations and any possible propagation of them are not understood as yet.

Comment #6:   Why the profiles are so structured? Does it mean /indicate in-situ generation in individual layers?

Response:   We do not believe that the vertical structures are due to generation in individual layers, because there are so strong vertical correlations of amplitudes and phases in the whole altitude regime in Fig.1. This is typical of all oscillation periods. It rather appears to indicate a global oscillation mode. In Part 1 of this paper (ACP 2021, Table 4) we have tentatively compared the structures to the zonal wind direction and vertical temperature gradient. Details are, however, complicated and presently under analysis (see #4 above).

Comment # 7:  "Line 676: Could you specify typical time scales? This might be important for long-term trend determination."

Response:  Time scales of CPT increases/decreases may be estimated from Fig.9 to be on the order of 30 years. This number has been added to Section V, last paragraph.

Comment #8:  "References :  Steiner et al.(2020) is missing in the list of references. Pisoft et al.- is it in press (list of references) or (2021) in the text ?

Response:  Steiner et al. (2020) has been added to the list of references. Thank you!

      Pisoft:  Correct citation is: Pisoft et al., (2021). List of reference has been changed accordingly. Thank you!

Comment #9:  "Wording and misprints: 301:"16" should be 160"!

Response:  Misprint has been corrected. Thank you!

**Citation**: https://doi.org/10.5194/acp-2022-677-AC1

**RC2**: ['Reply on AC1'](), Anonymous Referee #1, 22 Nov 2022

Reviewer #1.

I am satisfied with your response to my comments. I can recommend publication of the paper.

**Citation**: https://doi.org/10.5194/acp-2022-677-RC2

**RC3**: ['Comment on acp-2022-677'](), Anonymous Referee #2, 17 Dec 2022

I have only a couple minor suggestions for the improvement of this manuscript:

1) Since there are no staistical significance computations in the current manuscript the previous 2021 paper should be cited as containing these significane results where appropriate.

2) I would argue that land surface forcing remains a potential source for these long-period oscillations. This would mere require a teleconnected response to land surface forcing permitting a global response to local land surface variations. This should be noted unless the authors can argue that such a teleconnection is impossible.

3) A 1984 paper in J. Atmos Sci. by Ed Lorenz. depicted long period temperature variations in a simple moist model of the atmopsphere. s it worth mentionioning variations in the hydrologic cycle as a possible 'cause ' of these low frequency oscillations?

**Citation**: https://doi.org/10.5194/acp-2022-677-RC3

- **AC2**: ['Reply on RC3'](), Dirk Offermann, 22 Dec 2022

Response to comments of Referee 2 (RC3)

We thank the Referee for his helpful comments and suggestions! Corresponding changes to our text are indicated in red.

Comment #1

"Since there are no statistical significance computations in the current manuscript the previous 2021 paper should be cited as containing these significance results where appropriate".

Response:

We have added a corresponding remark in Section II, second paragraph: "The statistical significance of the period values presented in this paper has been analyzed in the preceeding paper of Offermann et al. (2021, Section 3.2)."

Comment 2:

"I would argue that land surface forcing remains a potential source for these long-period oscillations. This would mere require a teleconnected responnse to land surface forcing permitting a global response to local land surface variations. This should be noted unless the authors can argue that such a teleconnection is impossible."

Response:

Thank you for insisting on this point! It led us to a fresh view on our data!

Our basic idea was to compare pairs of locations/situations that differed in vegetation due to season, latitude, etc. We hoped to see differences that might be attributed to land-atmosphere interaction. We looked for such differences in oscillation periods and amplitudes, and did this in a large altitude range. We did not find significant differences. A recent paper by Desai et

al., (2022) suggested to look at the lowermost troposphere, what we did. Here we finally found the expected differences,- and only here!

Our present view is therefore an atmosphere with self-excited oscillations at almost all altitudes, and possibly externally excited oscillations at the lowermost levels.

We have re-formulated our paper accordingly, and indicated the changes in red in the text. The corresponding part of our Section V " Summary and Conclusions" was thus complemented as follows:

"The periods obtained for the contrasting cases are all found very similar.

The same holds for the vertical profiles (up to the mesopause) of the oscillation amplitudes at most altitudes. It is therefore concluded that the oscillations most likely are internally excited in the atmosphere.

There is, however, one exemption. Land-atmosphere interactions should mainly occur in the lowermost atmophere (boundary layer). We therefore considered especially the lowest atmospheric levels. Here, indeed, the vertical amplitude profiles showed peculiar structures that we tentatively attribute to land-atmosphere interactions. The peculiarities quickly disapper at higher altitudes. Hence we obtain the preliminry picture of self-excited oscillations in the upper atmosphere, and possible land surface excitation at the lowest levels."

Comment 3:

"A 1984 paper in J.Atmos.Sci. by Ed Lorenz depicted long period temperature variations in a simple moist model of the atmosphere. Is it worth mentioning variations in the hydrological cycle as a possible "cause" of these low frequency oscillations?"

Response:

The Lorenz paper appears not immediately helpful as its time scale is about one year only. The hydrological cycle is not believed to be a candidate for the oscillation excitation because the sea surface temperatures have been kept constant in our model runs ( as mentioned in this and the previous paper).